# Consumer Motives for Choosing Fruit and Cereal Bars—Differences Due to Consumer Lifestyles, Attitudes toward the Product, and Expectations

**DOI:** 10.3390/nu14132710

**Published:** 2022-06-29

**Authors:** Małgorzata Kosicka-Gębska, Marzena Jeżewska-Zychowicz, Jerzy Gębski, Marta Sajdakowska, Katarzyna Niewiadomska, Robert Nicewicz

**Affiliations:** Department of Food Market and Consumer Research, Institute of Human Nutrition Sciences, Warsaw University of Life Sciences (SGGW-WULS), Nowoursynowska 159C, 02-776 Warsaw, Poland; marzena_jezewska_zychowicz@sggw.edu.pl (M.J.-Z.); jerzy_gebski@sggw.edu.pl (J.G.); marta_sajdakowska@sggw.edu.pl (M.S.); katarzynakwiecin@gmail.com (K.N.); s197097@sggw.edu.pl (R.N.)

**Keywords:** fruit and cereal bars, sugar, consumer segmentation, health threats, lifestyle

## Abstract

Fruit and cereal bars are the response to the changing needs of consumers seeking health-promoting and convenient products. A cross-sectional study was conducted using the CAWI (Computer-Assisted Web Interview) method, with 1034 respondents consuming products of this kind. The aims of the study were (1) to identify consumer segments based on the importance they attached to the selected attributes of fruit and cereal bars and (2) to characterize the identified segments in terms of frequency and reasons for the consumption of fruit and cereal bars, views on their impact on health, and consumer behavior related to the selected lifestyle elements. Five distinct consumer clusters were identified. Involved and Health-oriented were more likely to consume bars, perceiving them as nutritious products, with a positive impact on health. Frugal and Visual consumed fruit and cereal bars the least frequently. They paid little attention to choosing healthier products in daily diet and physical activity. The Information seekers consumed bars to reduce stress and to improve their mood.

## 1. Introduction

Nowadays, consumers pay increasing attention to the importance of a healthy diet [1], and this trend towards a healthy and active lifestyle and being “fit” generates an interest in foods that help maintain well-being and fitness [2]. However, the consumption of some products that are harmful to health remains high or even increases, and among these products is confectionary.

Confectionery tends to be high in energy due to its high sucrose and fat contents. Studies show that the increasing proportion of sugars in the diet is a health problem. Excessive consumption of free sugar is associated with tooth decay, type 2 diabetes, cardiovascular disease, liver steatosis, metabolic syndrome, and obesity [3,4,5]. It may also contribute to changes in neuronal systems with alterations in the processing of emotions, anxiety, and depression [6].

The consumption of confectionery is associated with many dilemmas. On the one hand, consumers are aware that they should limit or eliminate sweet products from their daily diet and, on the other hand, the habitual sweet taste makes consumers ingest them eagerly [7]. The sweet taste of products, as opposed to the bitter taste, is associated with safety and inspires confidence [8,9].

The consumption of sugar-containing products worldwide exceeds the recommended standards, especially among children and adolescents [10,11]. The reason for this situation is a greater availability of products with added sugars [12,13] and incorrect lifestyle habits particularly characteristic of developed countries [14].

The available data indicate that one in four global consumers has increased his/her confectionery consumption in recent years [15], which is also confirmed by other research [16,17,18]. The situation is similar in Poland. However, we still consume three times less sweet food than the inhabitants of other European countries, such as Germany [15]. In such situations, globally applied sugar reduction strategies have been targeted at product reformulation or the creation of new products that could provide fewer calories [18]. Product reformulation targeted at reducing sugars is one of many proposed policies to reduce sugar intake at the population level [16,17,19,20]. Nevertheless, changes in sugar intake are small [16]. Several public policies targeted at promoting healthier eating habits have been recommended, including taxes and subsidies, restrictions on food marketing, and the provision of information [17]. Confectionery manufacturers are intensively seeking innovative technological solutions by incorporating innovative additives with health-promoting properties into their products [16,19]. They use sustainable production processes such as organic, fair-trade, and other production methods that aim to reduce energy consumption and environmental waste emissions throughout the food and packaging chain [20]. Fruit and cereal bars may be manufacturers’ answer to the changing needs of consumers looking for food products with health-promoting ingredients and new flavors in the confectionery market.

Because changing existing eating habits often requires limits on sweets and snacking, which is difficult to achieve, products such as fruit and cereal bars can help to reduce the consumption of sweets and to develop a new snacking habits that are important because of their convenience [21]. However, the acceptance of reformulated confectionary may be questionable. Consumers tend to be more accepting of healthier carriers in functional foods [22,23,24]. However, some previous research has shown that the reformulation of confectionery products is also accepted by consumers [16,18]. Cereal bars are perceived as a healthier alternative to chocolate bars [25,26]. When buying confectionery, Poles increasingly pay attention to its health benefits. Thus, for example, chocolates with a high cocoa content or cereal bars with other pro-health ingredients are growing in popularity [27].

The availability of cereal bars has increased the demand for them. They are now consumed worldwide as lower-calorie products [28]. They are consumed by people on a diet, those with health problems or consumers looking for a quick snack [29,30,31]. There is a growing popularity of cereal bars, which, being originally typical breakfast products, have become a desirable dietary component suitable for consumption at any time and under any circumstance [32].

Previous research has mainly focused on the production technology and sensory evaluation of cereal bars [28,29,30,31]. It is known that this ultra-processed food has received criticism from numerous consumer groups, the media, the public, and policy makers [33], which encourages the studies of their nutritional properties [34,35]. Obtained findings have shown that cereal bars have the potential to be a healthier and more natural alternative to chocolate bars [33]. However, consumer interest in these products has not been studied intensively. Hence, the reasons for their consumption, consumers’ perceptions of their health consequences, and the relationship between fruit and cereal bar consumption and consumers’ lifestyles, diets, physical activity levels, and leisure time preferences have not yet been fully recognized.

In a survey, an attempt was made to determine the characteristics of consumers of fruit and cereal bars, why they consume them, and whether they have concerns and expectations regarding this product category. Therefore, the aims of the study were (1) to identify consumer segments based on the importance they attach to selected attributes of fruit and cereals bars and (2) to characterize the identified segments in terms of frequency and reasons for consumption of fruit and cereal bars, views concerning their influence on health, as well as consumer behavior related to the selected elements of lifestyle.

## 2. Materials and Methods

### 2.1. Ethical Approval

The Ethics Committee of the Faculty of Human Nutrition and Consumer Science, Warsaw University of Life Sciences (SGGW) appointed on the basis of Regulation No. 27 of the SGGW Rector of 5 May 2016, approved the protocol of the analysis of the behavior of Polish consumers in the sweets market and of the determinants of consumer acceptance of innovative changes aimed at counteracting obesity 25 June 2018, Resolution No. 30/2018, as consistent with the guidelines laid down in the Declaration of Helsinki. Informed consent was provided by participants.

### 2.2. Data Collection Process

A nationwide quantitative survey on consumer behavior towards fruit and cereal bars was conducted by a professional market research agency, ARC Rynek i Opinia, in 2020. The survey sample included 1034 people aged 18 or older. Participants of the survey were recruited from an online panel (ePanel) of 65,000 people. The computer-assisted web interview method (CAWI) was used to collect the data. The sample selection in the study was purposive. People who are consumers of fruit and cereal bars were invited to fill in the questionnaire. Due to the lack of statistical data about the distribution of this feature in the population of Polish consumers, no quotas were imposed on individual demographic variables (gender, age, education, place of residence). To prevent a possible “bias” in the sample, basic demographic data were also collected on people who are not bar consumers and were excluded from the samples on the basis of their answers to the question about the consumption (so-called screenouts). Analysis of these data, together with category user data, did not reveal significant deviations from the population profile in the selected demographic categories.

The study was preceded by a pilot study through personal interviews with 35 respondents to identify and eliminate potential problems related to understanding the questions.

### 2.3. Questionnaire

#### 2.3.1. Consumer Behavior towards Fruit and Cereal Bars

The importance of different motives for choosing fruit and cereal bars in the decision to eat them was assessed using the question “Please specify which of the following factors you pay attention to when choosing fruit and cereal bars”. A 5-point scale was used, where 1 meant “I never pay attention” and 5 meant “I always pay attention”. Seventeen factors were taken into account in the study, i.e., price, taste, product quality, brand/manufacturer, product appearance, package appearance, package size, friends’ opinion, advertising, the nutritional value of the product, the caloric value of the product, ingredients enriching the product, shelf life, promotion, place of production, habit, and organic origin.

To assess the frequency of consumption of fruit and cereal bars, the question “How often do you consume fruit and cereal bars?” was used with the following responses: (1) every day, (2) 1–2 times a week, (3) several times a month, or (4) several times a year. In the question about the reasons for consuming fruit and cereal bars (“What is the main reason for reaching for fruit and cereal bars?”) the following were included: (1) stress, (2) improvement of mood, (3) perceived hunger, (4) desire to eat something sweet, (5) desire to eat a nutritious product, (6) desire to take care of one’s health, and (7) other answers. The respondent could select only one answer.

Three questions were used to explore views on the health effects of fruit and cereal bars. The first concerned the respondents’ tendency to limit their consumption (“Do you try to limit your consumption of fruit and cereal bars?”). Consumers could choose the following answers: (1) yes, I still limit their consumption, (2) yes, I limit it but I still give in to cravings, (3) I do not limit it but I would like to do so, and (4) I do not pay attention to it. The second question concerned the respondents’ opinion on the influence of fruit and cereal bars on their health (Please tell us how fruit and cereal bars influence your health?”). Respondents could select the following response categories: (1) negative impact, (2) positive impact, and (3) no impact at all. The third question asked respondents’ opinions about their health concerns related to the consumption of the bars (“Which of the following health consequences are you worried about after consuming fruit and cereal bars?”), with consequences such as: (1) food allergies, (2) weight gain/obesity, (3) learning bad eating habits, (4) tooth decay, (5) diabetes, (6) skin problems, and (7) digestive problems.

Sixteen innovative recipe modifications were used to assess consumers’ expectations of recipe changes that could encourage them to consume fruit and cereal bars in the future. Respondents rated this on a 5-point scale, where 1—“the modification is of very little importance to me” and 5—“the modification is of very high importance to me”.

#### 2.3.2. Lifestyle of Respondents

Lifestyle questions included the number of meals consumed per day, preferred leisure activities, assessment of physical activity during work and leisure time, number of hours per day devoted to sleep, and smoking. The Kompan PAN questionnaire [36] was used to develop the questions. The question “How do you most like to spend your leisure time?” was added. From seven answers (sitting in front of the computer, watching TV, reading books/newspapers, listening to music, doing physical activities, doing housework, lying/sleeping), respondents chose one.

#### 2.3.3. Socio-Demographic Characteristics of the Respondents

The metric part included the socio–demographic characteristics of the subjects such as gender and age, place of residence, education, occupational situation opinion on income, and weight and height to calculate Body Mass Index (BMI). The answers to these questions are presented in Table 1.

### 2.4. Statistical Analysis

Qualitative variables are presented as percentages (%), and the Chi-square test was used to verify the differences between them. Factor Analysis (FA) was used to extract and identify the determinants of the choice of fruit and cereal bars. The factors were rotated by an orthogonal transformation (varimax rotation). The number of factors was determined by a scatter plot and the eigenvalue of the factor (above 1). The correctness of factor selection was confirmed using the Kaiser–Meyer–Olkin (KMO) test.

The factors obtained by factor analysis explained 62.85% of the total variation. The qualification of particular factors was performed based on the minimum value of factor loadings estimated at the 0.5 level. Analysis of statements’ reliability in a given question was performed using Cronbach’s coefficient alpha. The obtained value of Cronbach’s coefficient alpha = 0.803 confirmed the right choice of questions for factor analysis (Principal Component Analysis—PCA).

The resulting factors, for the ease of interpretation, were normalized using the “range” method (subtract the minimum and divide by SD) and then divided into quintiles.

The four new variables (factors) were the material for cluster analysis, which was carried out using the hierarchical method, which did not produce the desired result. The k-means method was then used to separate the clusters with the initial means coming from the hierarchical method. The validity of the extracted clusters was statistically confirmed by CCC (Cubic Clustering Criteria) and pseudo-F2 statistics.

The profiling of the clusters was carried out using the following sets of variables: frequency and reasons for consumption of fruit and cereal bars, views on their impact on health, attitudes towards product novelties and expected changes in the recipe for a new fruit and cereal bar, lifestyle of the respondents, and demographic characteristics.

One-way analysis of variance (ANOVA) with the post-hoc Waller–Duncan K-ratio t-test and Chi-square test were used to identify significant differences between the clusters. Significance was set at *p* < 0.05 for all analyses [37,38]. The statistical package SAS 9.4 was used for statistical analysis [39,40].

## 3. Results

### 3.1. Characteristics of the Study Sample

The socio–demographic characteristics of the study sample including gender, age, place of residence, education, professional situation, a subjective evaluation of the financial situation, and BMI are presented in Table 1. Most of the respondents were women (70%), people aged 25–39 years, with higher education, working full-time, and having a normal body weight. The fewest respondents were older than 55 years. More than half of the respondents had a university education, and more than 66% of the respondents had a full-time job. More than half of the respondents had a normal body weight, and 1/3 were overweight or obese.

### 3.2. Socio-Demographic Characteristics of the Identified Clusters

Out of 16 variables describing the reasons for choosing fruit and cereal bars, four factors were extracted by the applied factor analysis. Variables with a factor loading of at least 0.5 were included in the factors (Table 2).

The clusters identified according to the motives regarding fruit and cereal bar selection (factors) are presented in Table 3.

In cluster 1 (Frugal) the highest mean value was achieved by financial motives, in cluster 2 (Visual) by visual motives, in cluster 3 (Information seekers) by information motives, and in cluster 4 (Health-oriented), health-oriented motives. However, in cluster 5 (Involved), all of these motives were assessed as important (Table 3). Respondents paid more attention to financial motives, such as price or promotion. In Cluster 2, visual motives (the look of the product and packaging) were important. Cluster 3 paid attention to information motives, i.e., advertising, opinion of friends, or information that the product was manufactured in Poland. For Cluster 4, health-promoting motives were important, which included paying attention to the nutritional and calorific value, enriching ingredients, quality, or the ecological origin of the product. Cluster 5 was respondents who paid equal attention to all factors.

The socio–demographic characteristics of the identified clusters are presented in Table 4.

In cluster 1 (Frugal, 24.4%), most people were aged 18–24 and the fewest people were aged 25–39. The fewest people were employed, while the most people were learning or studying. Cluster 2 (Visual, 16.0%) had the most people aged 55–65 and residents of large cities. Cluster 3 (Information seekers, 20.1%) had the highest number of men, inhabitants of rural areas and smaller towns, and the lowest number of people with primary and vocational education, as well as pensioners. Cluster 4 (Health-oriented, 22.2%) had the highest number of women, people aged 25–39, and people with higher education. Cluster 5 (Involved, 16.3%) was characterized by the highest share of respondents aged 40–54, with secondary education, working part-time. In this cluster, there were the fewest respondents aged 18–24 and with a higher level of education.

### 3.3. Reasons for and Frequency of the Consumption of Fruit and Cereal Bars

Almost half of the respondents consumed fruit and cereal bars 1–2 times a week. Every 10th consumer consumed these products daily. Fruit and cereal bars were most frequently consumed by those in the Involved cluster, followed by Health-oriented and Information seekers. Frugal and Visual respondents consumed fruit and cereal bars with the lowest frequency (Table 5).

More than half of the study group did not tend to reduce the consumption of fruit and cereal bars. Every 3rd respondent tried to limit their consumption but sometimes gave in to cravings for something sweet. Most Information seekers, followed by Involved respondents, declared a tendency to limit their consumption of fruit and cereal bars. In contrast, the greatest number of Frugal, Visual, and Health-oriented respondents did not restrict the consumption of fruit and cereal bars (Table 5).

Almost half of the study group (48.2%) stated that eating fruit and cereal bars had a positive effect on their health. Such responses were characteristic of Health-oriented and Involved respondents. Only about 7% of respondents perceived a negative effect of fruit and cereal bars on their health, and among them were the most Frugal and the least Involved respondents. Frugal respondents were the most likely to hold this view, while Involved respondents were the least likely. Although about 45% of respondents were unable to assess this effect, a lot of people reported the negative consequences of eating candy bars such as weight gain (42.4%), tooth decay (32.6%), and diabetes (27.5%). Fewer Visual respondents and Information seekers than others were concerned about weight gain. On the other hand, most Information seekers were concerned about skin and digestive problems (Table 5).

Consumers of fruit and cereal bars ate them primarily due to the desire to eat something sweet or to eat a nutritious product. Stress was indicated by the fewest number of people as a factor for bar consumption. Most of the Frugal respondents and Information seekers consumed fruit and cereal bars because of hunger. Furthermore, more than 2/5 of Frugal respondents but also half of the Information seekers indicated the desire for something sweet as the reason for eating the bars. Most Information seekers consumed candy bars because of stress and to improve their self-esteem. The desire to eat a nutritious product and to take care of one’s health was indicated by the highest number of people in the Health-oriented cluster compared to other clusters, but the nutritional values of the product also prompted people in the Involved cluster to eat a chocolate bar (Table 5).

### 3.4. Expectations Concerning the Reformulation of Fruit and Cereal Bars

The changes expected in the fruit and cereal bars were mainly a reduction in the sugar content (mean score = 4.07) and the addition of fruit (mean score = 4.05). Minerals and vitamins, omega-3 fatty acids and fiber enrichment also ranked high. The least expected change was the addition of beta-glucan (mean score = 3.18), wheat germ, (mean score = 3.56), and vegetables (mean score = 3.62)—(Table 6).

Reported expectations regarding reformulation differed among clusters. Recipe changes in innovative fruit and cereal bars were expected the most by persons belonging to the Involved and Health-oriented clusters. Consumers from the Involved cluster expected changes in the form of enrichment of bars with omega-3 acids, vitamins and minerals, fiber, and the addition of fruit. Health-oriented consumers were interested in lowering the sugar content, enrichment in vitamins and minerals, and lowering the salt and fat contents (Table 6).

### 3.5. Lifestyle Factors as Determinants of Fruit and Cereal Bar Choice

Almost 90% of respondents eating fruit and cereal bars declared that they consumed between three and five meals per day (Table 7). Cereal products, meat and meat products, and vegetables were the products consumed most frequently by people eating fruit and cereal bars during the day. Every second respondent did physical activities as well as housework (e.g., cleaning, cooking). Respondents rarely spent their free time passively. More than 70% of respondents who declared to consume fruit and cereal bars did not smoke cigarettes.

Frugal people consumed cereal products and sweets most often and fruit least often. They spent their free time passively, mainly reading books and newspapers. At the same time, more Frugal people declared low physical activity during work or school compared to others. When evaluating their physical activity during work or school, they were more likely than other clusters to indicate that it was low. Visual people consumed the most meals per day, i.e., six or more meals. They consumed fruit least often and milk and dairy products most often. In their leisure time, in comparison with other clusters, they were the least engaged in physical activity and the most in doing housework and lying or sleeping. Physical activity in leisure time was most often assessed as low.

Information seekers are people who, compared to others, declared eating fewer than three meals per day. They consumed cereal products the least often and fruit the most often. They spent their free time sitting in front of a computer, watching TV, and listening to music. They were least likely to do housework. Health-oriented people ate between two and five meals per day. They ate vegetables most often and meat and meat products and sweets least often. They spent their free time mainly doing physical activities. They were least likely to sit at a computer. They declared moderate and high physical activity more often than others. In this cluster, there was the largest number of people who did not smoke, but at the same time, more than one-fifth of the representatives of this cluster smoked cigarettes.

Respondents from the Involved cluster were distinguished by the fact that they consumed milk and dairy products least frequently and meat and meat products most frequently. Involved were the people with the highest percentage who smoked cigarettes. Involved people rated their physical activity during work as moderate to average in comparison with other clusters.

## 4. Discussion

Our research identified five clusters considering the similarity of the motives for choosing the fruit and cereal bars, i.e., Frugal (24.4%), Health-oriented (22.2%), Information-seekers (20.1%), Visual (16.0%), and Involved (16.2%). Clusters showed differences after taking into account such socio–demographic characteristics as gender, age, education level, and region, but also due to selected elements of lifestyle and expectations regarding a new product that could appear on the market.

Fruit and cereal bars were significantly more likely to be consumed by consumers in the Health-oriented and Involved clusters, primarily due to the desire to eat a nutritious product and to take care of their health. Although consumers tend to be more accepting of healthier carriers in functional foods [22,24], it appeared that the presence of cereal and fruit in the bars may be more meaningful to these people than the confectionary product itself. Among these people, there were relatively few “difficult to say” opinions on the healthy effect of eating fruit and cereal bars and few negative opinions, which may indicate a strong belief in the health benefits. Such an interpretation is supported by other studies that show that cereal bars are often perceived as a healthier alternative to, for instance, chocolate bars [27,28]. However, research on the nutritional quality of cereal bars has shown a high sugar content, but also valuable amounts of fiber in these products [34,35]. Thus, the higher acceptance of bars in the two clusters may be due to the higher fiber content, the presence of fruits perceived as healthy, and also the lower content of saturated fat compared to chocolate bars, which are also perceived as highly processed, high in artificial additives, and thus unnatural [41].

Those for whom health motives were primarily important when choosing fruit and cereal bars (Health-oriented) were mainly younger women, well-educated, and living in urban areas, which reflects the characteristics of people for whom health is important in a food choice [42,43], and for those displaying more correct eating behaviors [44,45]. Similar characteristics were represented by Involved people; however, in this cluster, there were more people with secondary education compared to Health-oriented people. Involved people ate fruit and cereal bars more frequently compared to Health-oriented people. Twice as many Involved as Health-oriented people indicated that they eat fruit and cereal bars daily. This may indicate that the fruit and cereal bars are treated by them similarly to sweets.

The lowest number of Visual and Frugal people consumed fruit and cereal bars on a daily basis, while at the same time, they were candy eaters. Thus, this may indicate a lack of acceptance of fruit and cereal bars as a substitute for sweets. The Frugal diet included a lot of cereal products, but less fruit compared to other clusters, which may explain the low interest in adding fruit to fruit and cereal bars. Thus, those groups should be addressed with a crop-based product, such as cereal bars with quinoa, with beneficial effects on the cardiovascular system [29]. The addition of fruit flour may be considered an alternative to fruit [46,47]. Both clusters were different in terms of age, as Frugal people were younger than Visual people. The older age of the Visual people may explain the differences in their diet compared to that of the Frugal people, i.e., low amount of vegetables and high amount of milk and milk products, with similar consumption of sweets, meat, and meat products.

Some motives for choosing fruit and cereal bars were reflected in the lifestyle characteristics of people who represented them. Health-oriented people were more physically active at leisure time, while Involved people were more physically active during work time. Previous studies have shown that people with a higher level of physical activity displayed healthier dietary choices [48,49], which was also confirmed among Health-oriented people. However, frequent consumption of fruit and cereal bars among the Involved people may be treated as an element of their lifestyle similarly to consumption of sweets, and at the same time, as a form of taking care of their health (high importance of a health motive). In turn, Frugal and Visual people were characterized by passive leisure time activities, but also low physical activity during work, which may favor the consumption of snacks, including sweets rather than fruit and cereal bars [50,51], as confirmed by the results.

The identified clusters differed in terms of the expected changes in the formulation of new fruit and cereal bars. Frugal, Visual, and Health-oriented people would like to see a reduction in the sugar content, and Health-oriented people would also like to a see a reduction in the salt content, which is in line with nutrition recommendations [52], whereas all clusters except for Health-oriented people expected the addition of fruit. This trend is consistent with the high interest in snacks and sweets made from fruit or with the addition of fruit and flavors [53]. Except for Frugal people and Information seekers, respondents expected the new bars to be fortified with vitamins and minerals. In turn, Involved people pointed out the enrichment with fiber and the use of organic ingredients in production. All expectations proved that consumers are aware of the role of particular food ingredients in the diet [54], both unfavorable (salt and sugar) and favorable (vitamins, minerals, fiber). The whole grain cereal used in the bars increases the fiber content, but also provides other important compounds such as vitamins and minerals [55], which are of great importance for health [56,57]. Thus, cereal bars with additional ingredients can partly replace confectionery in the diet [58,59,60]. Modifying the composition of innovative bars may allow manufacturers to adjust the product to the needs of consumers with special dietary requirements, e.g., regarding gluten [61].

The findings of the study have shown that respondents did not see any reason to limit the consumption of fruit and cereal bars. This may be associated with the familiarity of these products, perceived as sweets with higher nutritional quality and usually a higher degree of naturalness compared to chocolate bars [33]. However, the nutritional quality of cereal bars varied greatly [35], and they are only slightly more natural than chocolate bars [62], which may be a reason to change the choice of bars in the future. In addition, for the consumer, a healthier fruit and cereal bar should still be a tasty bar, and at the same time, chocolate plays an important role in the consumer’s liking of cereal bars [63,64] so eliminating it from the product by replacement with fruit may not always be acceptable. Further research on the consumer acceptability of cereal bars with regard to their composition and also motives of choice is needed.

### Limitations of the Study

The data were collected among Polish adults; hence, the results cannot be generalized to other populations due to the differences associated with ethnicity or socioeconomic status. Another limitation of the study is that the study group was not representative of the Polish population as a whole. Moreover, we used purposive sampling in the study. Only consumers declaring to consume fruit and cereal bars participated in the survey. Due to the lack of statistical data on the consumption pattern of fruit and cereal bars in the population of Polish consumers, no quotas were imposed on the individual demographic variables. The authors believe that such limitations are compensated by the large study sample. However, generalizations and extrapolations at a broader level are not allowed under this procedure. Finally, the cross-sectional design of this study does not allow the identification of a causal relationship.

## 5. Conclusions

The results of the study showed that in general, the study participants were interested in fruit and cereal bar consumption. They believed that these products have a positive effect on health and did not want to limit the frequency of their eating; in particular, such beliefs were presented by the Health-oriented and Involved consumers. Frugal and Visual consumers consumed fruit and cereal bars with the least frequency yet were the most interested in their consumption. Hence, promotional activities for fruit and cereal bars should be targeted primarily at these consumers.

The obtained results may be useful for producers of fruit and cereal bars as they present characteristics of potential consumers of their products including their socioeconomic and demographic features, elements of their lifestyles, and changes in recipes, which they expect in bars. The differences in consumers’ expectations regarding the product characteristics and its recipe have shown that fruit and cereal bars cannot be perceived as a homogenous group of products, similar to their potential buyers.

## Figures and Tables

**Table 1 nutrients-14-02710-t001:** Socio–demographic Characteristics of the study sample.

Variables		Total Sample	
		*n*	%
Total sample		1034	100.0
Gender	Female	716	69.25
	Male	318	30.75
Age	18–24 years	145	14.02
	25–39 years	556	53.77
	40–54 years	283	27.37
	55–65 years	50	4.84
Place of residence	Countryside	185	17.89
	City up to 100,000 inhabitants	377	36.46
	City with more than 100,000 inhabitants	472	45.65
Education	Primary	16	1.55
	Elementary vocational	76	7.35
	Secondary	411	39.75
	Higher	531	51.35
Professional situation	I work full time	689	66.63
	I have a part-time job	74	7.16
	I neither work nor study	103	9.96
	I learn/study and work at the same time	34	3.29
	I learn/study (and not work)	55	5.32
	I am self-employed	40	3.87
	I am retired	39	3.77
Income opinion	Is not sufficient at all	26	2.52
	Only allows us to meet our basic needs	127	12.28
	We can afford some, but not all expenses	572	55.32
	We can afford everything	227	21.95
	We can afford everything, plus we can save	82	7.93
BMI categories	Underweight	58	5.61
	Normal weight	586	56.67
	Overweight	284	27.47
	Obese	106	10.25

**Table 2 nutrients-14-02710-t002:** Factor analysis of the choice motives regarding fruit and cereal bars; varimax rotated factor loadings; and percentage of explained variance (*n* = 1034, Poland).

Reasons for Choosing Bars	Factor 1 The Health-Promoting Motives	Factor 2 The Visual Motives	Factor 3 The Information Motives	Factor 4 The Financial Motives
Nutritional value of the product	0.816			
Ingredients to enrich the product	0.803			
The calorific value of the product	0.790			
Quality of the product	0.644			
Organic origin of the product	0.571			
The appearance of the product		0.850		
The appearance of the packaging		0.829		
Brand/manufacturer		0.620		
Package size		0.609		
Advertising			0.742	
Friends’ opinion			0.694	
It was produced in Poland			0.519	
Price				0.821
Promotion				0.773
Flavor				
Best-before date				
Variance explained	33.96%	10.98%	10.61%	7.30%
Kaiser’s Measure of Sampling Adequacy: Overall MSA = 0.85				

**Table 3 nutrients-14-02710-t003:** Characteristics of the clusters identified according to the motives regarding fruit and cereal bar selection (factors), (mean; standard deviation, *n* = 1034, Poland).

Factors/Fruit and Cereal Bar Selection Motives	Total Sample (*n* = 1034)	SD	Cluster 1 Frugal (*n* = 252)	Cluster 2 Visual (*n* = 165)	Cluster 3 Information Seekers (*n* = 218)	Cluster 4 Health-Oriented (*n* = 230)	Cluster 5 Involved (*n* = 169)	*p*-Value
Factor 1—The health-promoting motives	3.02	1.44	2.60 ^c^	2.08 ^d^	1.79 ^e^	4.46 ^a^	4.17 ^b^	<0.0001
Factor 2—The visual motives	3.07	1.42	2.46 ^c^	4.18 ^b^	2.21 ^d^	2.35 ^c,d^	4.58 ^a^	<0.0001
Factor 3—The information motives	2.98	1.38	2.25 ^d^	1.65 ^e^	4.52 ^a^	2.59 ^c^	4.04 ^b^	<0.0001
Factor 4—The financial motives	3.00	1.36	4.50 ^a^	2.08 ^c^	2.17 ^c^	2.08 ^c^	3.98 ^b^	<0.0001

^a–e^ Means in the same line with the same letter are not significantly different; one-way ANOVA, *p* < 0.05.

**Table 4 nutrients-14-02710-t004:** Socio–demographic characteristics of the study sample according to clusters’ adherence (%, *n* = 1034, Poland).

Variables	Total Sample (*n* = 1034)	Cluster 1 Frugal (*n* = 252)	Cluster 2 Visual (*n* = 165)	Cluster 3 Information Seekers (*n* = 218)	Cluster 4 Health-Oriented (*n* = 230)	Cluster 5 Involved (*n* = 169)	*p*
Gender							
Female	69.25	73.40	70.30	55.50	77.83	68.05	<0.0001
Male	30.75	26.59	29.70	44.50	22.17	31.95	
Age							
18–24 years	14.02	22.22	16.97	12.39	8.70	8.28	0.0011
25–39 years	53.77	47.62	47.88	57.8	59.13	56.21	
40–54 years	27.37	26.19	27.88	27.06	26.52	30.18	
55–65 years	4.84	3.97	7.27	2.75	5.65	5.33	
Place of residence							
Countryside	17.89	17.46	17.58	21.10	15.22	18.34	0.0072
City up to 100,000 inhabitants	36.46	34.52	33.33	38.53	37.39	38.46	
City with more than 100,000 inhabitants	45.65	48.02	49.09	40.37	47.39	43.20	
Education							
Primary	1.55	1.59	2.42	1.83	0	2.37	0.0011
Vocational	7.35	3.97	6.06	12.84	4.35	10.65	
Secondary	39.75	42.46	41.82	34.86	36.52	44.38	
Higher	51.35	51.98	49.70	50.46	59.13	42.60	
Professional situation							
I work full time	66.63	58.73	66.06	66.51	73.48	69.82	0.0063
I have a part-time job	7.16	6.35	4.85	8.26	6.52	10.06	
I neither work nor study	9.96	11.51	10.91	11.01	7.83	8.28	
I learn/study and work at the same time	3.29	6.35	3.03	2.29	2.61	1.18	
I learn/study (and not work)	5.32	10.32	6.06	3.21	1.74	4.73	
I am self-employed, a business owner	3.87	3.97	4.24	3.67	4.78	2.37	
I am retired	3.77	2.78	4.85	5.05	3.04	3.55	

*n*—numbers of respondents; *p*-value—Chi-square test.

**Table 5 nutrients-14-02710-t005:** Frequency of consumption and opinions on the health effects of fruit and cereal bar consumption according to the clusters’ adherence (%, *n* = 1034, Poland).

	Total Sample (*n* = 1034)	Cluster 1 Frugal (*n* = 252)	Cluster 2 Visual (*n* = 165)	Cluster 3 Information Seekers (*n* = 218)	Cluster 4 Health-Oriented (*n* = 230)	Cluster 5 Involved (*n* = 169)	*p*
Frequency of eating fruit and cereal bars							
Each day	10.64	5.16	7.27	11.47	10.87	20.71	<0.0001
1–2 times a week	46.91	39.29	41.82	48.62	54.78	50.30	<0.0001
Several times a month	28.81	36.11	34.55	27.52	24.78	19.52	<0.0001
Several times a year	13.64	19.44	16.36	12.39	9.57	9.47	<0.0001
Tendency to reduce the consumption of fruit and cereal bars							
Yes, I still limit my intake	6.09	6.74	5.45	7.34	4.35	6.51	<0.0001
Yes, but sometimes I give in to cravings	29.88	24.21	13.33	46.78	29.13	33.73	<0.0001
No, but I would like to cut down on my intake	8.90	7.94	3.64	12.39	9.13	10.65	<0.0001
I do not pay attention to it	55.13	61.11	77.58	33.49	57.39	49.11	<0.0001
Opinions on the healthy effect of eating fruit and cereal bars							
Negative	6.29	9.92	5.45	5.05	6.09	3.55	<0.0001
Positive	48.16	38.49	46.67	42.20	56.09	60.95	<0.0001
Difficult to say	45.55	51.59	47.88	52.75	38.82	35.50	<0.0001
Opinions on the health consequences of eating fruit and cereals bars							
Food allergies	11.41	9.92	6.67	15.14	12.61	11.83	0.1083
Weight gain	42.36	48.81	37.58	34.4	44.78	44.38	0.0151
Learning unhealthy eating habits	21.95	24.6	17.58	23.39	19.57	23.67	0.3799
Tooth decay	32.59	29.37	29.09	34.86	34.35	35.5	0.4627
Diabetes	27.47	28.97	20.61	29.36	29.57	26.63	0.2729
Skin problems	10.44	9.92	7.27	15.60	8.70	10.06	0.0459
Digestive problems	12.38	9.13	9.09	20.64	12.17	10.06	0.0009
Reasons for the consumption of fruit and cereal bars							
Stress	2.81	0.40	1.21	8.26	0.86	3.55	<0.0001
To improve mood	7.34	3.96	6.05	15.14	3.49	8.87	<0.0001
Hunger	15.86	19.84	16.37	18.81	12.17	10.65	<0.0001
Craving something sweet	37.14	42.86	50.3	28.44	32.61	33.14	<0.0001
Want to eat something nutritious	29.59	25.4	21.82	23.39	40.00	37.28	<0.0001
To take care of my health	6.58	6.75	3.64	5.50	10.00	5.92	<0.0001
Other reason	0.68	0.79	0.61	0.46	0.87	0.59	<0.0001

Test of independence: Chi^2^. Statistically significant (*p* < 0.05); *n*—number of respondents.

**Table 6 nutrients-14-02710-t006:** Expected changes in fruit and cereal bars according to the clusters’ adherence (%, *n* = 1034, Poland).

Expected Changes	Total Sample (*n* = 1034)	Cluster 1 Frugal (*n* = 252)	Cluster 2 Visual (*n* = 165)	Cluster 3 Information Seekers (*n* = 218)	Cluster 4 Health-Oriented (*n* = 230)	Cluster 5 Involved (*n* = 169)	*p*
Enrichment with essential fatty acids (EFAs)	3.52	3.32 ^c^	3.17 ^c^	3.17 ^c^	3.79 ^b^	4.26 ^a^	<0.0001
Enrichment with vitamins and minerals	3.97	3.90 ^b^	3.73 ^c^	3.41 ^d^	4.36 ^a^	4.48 ^a^	<0.0001
Reduction of the fat content	3.90	3.94 ^b^	3.52 ^c^	3.37 ^c^	4.27 ^a^	4.37 ^a^	<0.0001
Reduction of sugar	4.07	4.19 ^b^	3.71 ^c^	3.45 ^d^	4.51 ^a^	4.45 ^a^	<0.0001
Reduction of salt	3.89	3.88 b	3.51 c	3.44 ^c^	4.28 ^a^	4.34 ^a^	<0.0001
Reduction of cholesterol	3.88	3.79 ^c^	3.60 ^d^	3.51 ^d^	4.15 ^b^	4.37 ^a^	<0.0001
Enrichment with protein	3.71	3.67 ^c^	3.48 ^d^	3.32 ^d^	3.89 ^b^	4.28 ^a^	<0.0001
Removing ingredients that cause allergies and sensitivities	3.63	3.50 ^c^	3.31 ^d^	3.25 ^d^	3.85 ^b^	4.33 ^a^	<0.0001
Enrichment with fiber	3.92	3.89 ^c^	3.68 ^d^	3.36 ^e^	4.24 ^b^	4.47 ^a^	<0.0001
Enrichment with omega-3 fatty acids	3.96	3.95 ^c^	3.71 ^d^	3.41 ^e^	4.25 ^b^	4.53 ^a^	<0.0001
Addition of fruit	4.05	4.18 ^b^	3.92 ^c^	3.53 ^d^	4.18 ^b^	4.46 ^a^	<0.0001
Addition of vegetables	3.62	3.56 ^c^	3.29 ^d^	3.33 ^d^	3.78 ^b^	4.16 ^a^	<0.0001
Addition of wheat germ	3.56	3.32 ^c^	3.30 ^c^	3.39 ^c^	3.69 ^b^	4.16 ^a^	<0.0001
Addition of beta-glucan	3.18	2.87 ^c^	2.75 ^c^	3.26 ^b^	3.23 ^b^	3.88 ^a^	<0.0001
Addition of chia seed/flaxseed	3.72	3.63 ^c^	3.39 ^d^	3.40 ^d^	3.91 ^b^	4.35 ^a^	<0.0001
Use of organic ingredients	3.87	3.72 ^c^	3.59 ^c,d^	3.42 ^d^	4.19 ^b^	4.47 ^a^	<0.0001

^a–e^ Means with the same letter are not significantly different; ANOVA with the post-hoc Waller–Duncan K-ratio *t*-test.

**Table 7 nutrients-14-02710-t007:** Profile of clusters in terms of lifestyle factors (%, *n* = 1034, Poland).

Lifestyle Factors	Total Sample (*n* = 1034)	Cluster 1 Frugal (*n* = 252)	Cluster 2 Visual (*n* = 165)	Cluster 3 Information Seekers (*n* = 218)	Cluster 4 Health-Oriented (*n* = 230)	Cluster 5 Involved (*n* = 169)	*p*
Number of meals per day							
Fewer than three	9.86	10.32	6.96	14.22	7.27	10.06	0.0497
Three to Five	87.14	87.7	89.13	82.57	90.31	86.39	
Six or more	3.00	1.98	3.91	3.21	2.42	3.55	
Group of products consumed most frequently during the day							
Cereal products	24.08	28.97	24.85	18.35	26.52	20.12	<0.0001
Vegetables	20.60	18.25	11.52	20.64	26.52	24.85	
Fruit	17.02	9.92	15.15	24.31	18.70	17.75	
Milk and milk products	13.83	15.08	21.21	11.93	11.30	10.65	
Meat and meat products	21.47	22.62	23.03	22.48	16.53	23.67	
Sweets	3.00	5.16	4.24	2.29	0.43	2.96	
The way you spend your free time							
I sit in front of the computer	13.54	13.89	15.15	15.60	9.13	14.79	<0.0001
I watch TV	16.34	11.51	16.97	27.06	13.04	13.61	
I read books/newspapers/magazines	19.44	23.81	21.82	13.30	20.44	17.16	
I listen to music	8.23	7.14	9.08	10.55	6.09	8.88	
I do physical activities	20.31	19.84	11.52	18.35	30.43	18.34	
I do household chores (e.g., cleaning, cooking etc.)	20.21	21.03	21.82	13.76	20.44	25.44	
I lie down/sleep	1.93	2.78	3.64	1.38	0.43	1.78	
Assessment of physical activity during work or school							
Low *	42.55	48.81	48.48	38.99	41.30	33.73	0.0388
Moderate	43.43	38.49	35.76	48.62	45.22	49.11	
High	14.02	12.7	15.76	12.39	13.48	17.16	
Assessment of leisure-time physical activity							
Low **	30.07	34.13	39.39	27.98	20.87	30.18	0.0065
Moderate	56.58	50.79	51.52	58.26	63.91	57.99	
High	13.35	15.08	9.09	13.76	15.22	11.83	
How many hours do you sleep a night?							
Less than 5 h	6.29	5.95	4.85	9.17	4.78	6.51	0.348
5–6 h	49.32	55.56	51.52	43.58	48.70	46.15	
7–8 h	37.72	32.14	35.76	40.37	41.30	39.65	
More than 8 h	6.67	6.35	7.88	6.88	5.22	7.69	
Smoking							
Yes	29.79	29.13	33.94	29.36	24.21	35.5	0.0484
No	70.21	70.87	66.06	70.64	75.79	64.5	

* Low—more than 70% of the time in a sitting position; Moderate—about 50% of the time sitting and about 50% of the time moving; High—about 70% of the time in motion or hard physical work; ** Low—mostly sitting, watching TV, in front of a computer, reading newspapers and books, light housework, walking 1–2 h per week; Moderate—walking, cycling, gymnastics, gardening or other light physical activity 2–3 h per week; High—cycling, running, gardening, allotment work or other sporting recreational activities requiring physical exertion for more than 3 h per week.

## Data Availability

The data are the property of the Institute of Human Nutrition Sciences.

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
