# Peer review of "Consumer Motives for Choosing Fruit and Cereal Bars—Differences Due to Consumer Lifestyles, Attitudes toward the Product, and Expectations"

_nutrients, 2022, doi:10.3390/nu14132710_

Round 1
Reviewer 1 Report
My comments are provided below.
1 Abstract
- Line 14: It will be helpful to write CAWI in full since that is the first use.
- Line 26: reference to chocolate bars seems off as the focus has been on fruit and cereal bars with no prior reference made to chocolate bars.
2. Introduction
- Line 58-60: Could the authors further explain the statement made?
_ Line 63 and line 72: fruit and cereal bars, as well as fruit and vegetable bars, were used interchangeably, could the authors confirm which is the correct one? For the sake of consistency, can only one of these terms be used throughout the paper?
3. Statistical Analysis
- Line 176: ”…hierarchical method which did not produce the desired results”. Can this statement be further expatiated? What was the result obtained from the hierarchical cluster analysis and why was it not desirable?
4. Results
- Characteristics of the study sample (line 190-16) is this representative of the population of the country of study?
- Line 199 “socjo-demographic”
5. Discussion
Results of the subjective assessment of fruit and cereal bars (Table 8, line 270) show that most respondents answered: “difficult to say”. Can the authors discuss the implications of this result?
6. Conclusion
Line 476: “Fruit and cereal bars can be a healthier and more natural alternative to chocolate bars”. Indicating this in the conclusion seems unfitting as this was not investigated in the study and no reference to chocolate bars was made in the introduction, rather, the focus was on fruits and cereal bars being healthier alternatives to high-calorie, sugar-containing confectionaries in general without specific reference to chocolate bars.
Author Response
Reviewer 1
Thank you very much for your very valuable comments and for the time you dedicated to reviewing our article.
We have highlighted our responses in yellow in the manuscript.
1 Abstract
- Line 14: It will be helpful to write CAWI in full since that is the first use.
It has been done.
- Line 26: reference to chocolate bars seems off as the focus has been on fruit and cereal bars with no prior reference made to chocolate bars.
We apologize for our mistake. We have written: fruit and cereal bars.
- Introduction
- Line 58-60: Could the authors further explain the statement made?
In lines 58-60 we have included information extracted from the following article: de Avelar, M.H.M.; de Castilho Queiroz, G.; Efraim, P. Sustainable performance of cold-set gelation in the confectionery manufacturing and its effects on the perception of sensory quality of jelly candies. Clean. Eng. Technol. 2020, 1, 100005. Its authors demonstrate that the approach and production method in confectionery manufacturers are changing.
_ Line 63 and line 72: fruit and cereal bars, as well as fruit and vegetable bars, were used interchangeably, could the authors confirm which is the correct one? For the sake of consistency, can only one of these terms be used throughout the paper?
The available research focuses on cereal bars. Our study focuses on fruit and cereal bars, which are also the same product category with the addition of fruit. In preparing the literature review, we could only refer to the characteristics of cereal bars. Hence, these two names appear in the text.
- Statistical Analysis
- Line 176: ”…hierarchical method which did not produce the desired results”. Can this statement be further expatiated? What was the result obtained from the hierarchical cluster analysis and why was it not desirable?
The hierarchical method failed to produce a positive CCC (Cubic Clustering Criteria) statistic, indicating insufficient separation of clusters. For the k-means method, this statistic had a local maximum for 5 clusters (8.67).
- Results
- Characteristics of the study sample (line 190-16) is this representative of the population of the country of study?
In our research project, we were to reach only people who consume fruit and cereal bars. A nationwide quantitative study on consumer behavior towards fruit and cereal bars was conducted by professional market research agency ARC Rynek i Opinia in 2020. The research sample included 1,034 people aged 18 years and older. Survey participants were recruited from an online panel (ePanel) of 65,000 people. The study sample is not representative of our nation's population. Due to the lack of the data on the distribution of this characteristic in the population of Polish consumers, no limits were imposed on individual demographic variables (gender, age, education, place of residence).
- Line 199 “socjo-demographic”
Thank you for your comment. We have changed the title of Table 1. We have deleted "socio-demographic".
- Discussion
Results of the subjective assessment of fruit and cereal bars (Table 8, line 270) show that most respondents answered: “difficult to say”. Can the authors discuss the implications of this result?
As Reviewer 3 noted, the data from Table 5-9 are now presented in a single table - Table 5. In response to your comment, the survey sample included only respondents who consumed fruit and cereal bars. That is, these products were familiar to them. The "difficult to say" response obtained (45.55%) may indicate that these respondents eat fruit and cereal bars but are not sure about their positive or negative health effects. We did not study it, but we suppose that this group of respondents, eat fruit and cereal bars because of their taste and convenience rather than health effects.
- Conclusion
Line 476: “Fruit and cereal bars can be a healthier and more natural alternative to chocolate bars”. Indicating this in the conclusion seems unfitting as this was not investigated in the study and no reference to chocolate bars was made in the introduction, rather, the focus was on fruits and cereal bars being healthier alternatives to high-calorie, sugar-containing confectionaries in general without specific reference to chocolate bars.
The Discussion and Conclusions have been modified following comments from all reviewers.
Thank you for this comment. The unfortunate sentence has been removed. It was not relevant to the results of our study.
Thank you very much for all your comments. We appreciate them very much. We hope that we were able to improve the article according to your suggestions. Thank you for taking the time to read and evaluate the text.

Reviewer 2 Report
The authors explore an interesting topic, I enjoyed read the paper which is well written and organized. The empirical analysis is thoroughly performed and results that are properly discussed.
I would suggest the authors to expand results discussion using updated literature, preferably as well as to provide more background information justifying their decision to employ as a case study Polish consumers, as well as fruit and cereal bars products
Author Response
Reviewer 2
Thank you very much for your very valuable comments and for the time you dedicated to reviewing our article.
The authors explore an interesting topic, I enjoyed read the paper which is well written and organized. The empirical analysis is thoroughly performed and results that are properly discussed.
Thank you for your comments. We appreciate them very much.
I would suggest the authors to expand results discussion using updated literature, preferably as well as to provide more background information justifying their decision to employ as a case study Polish consumers, as well as fruit and cereal bars products.
As suggested by the reviewer we have expanded the discussion of obtained results using updated literature. We have also provided more background information justifying our decision to choose Polish consumers and the products i.e. fruit and cereal bars products.
The Innova Market Insights data report indicates that one in four global consumers has increased their confectionery consumption in recent years. The data also shows that the global confectionery market has grown by up to 15 percent. Poles like confectionery and also consume more and more of it every year. However, they still consume three times fewer sweets than inhabitants of other European countries - e.g. Germany. As in other countries, Polish consumers are interested in 'bite-size' sweets (small snacks in smaller packages), as well as special sweets with reduced calories, which confirms the right choice of the research topic, and fruit and cereal bars as products increasingly bought by people taking care of their health and physical activity, which was also confirmed in our study (Rynek słodyczy w Polsce rośnie. Wyceniany jest na ponad 12,5 mld zł, https://finanse.wp.pl/rynek-slodyczy-w-polsce-rosnie-wyceniany-jest-na-ponad-125-mld-zl-6457640241543297a).
In addition, in Poland, the confectionery market continues a trend that has been present in many countries for several years, whereby desirable products are characterized by high quality and the origin of ingredients. The sought-after products are chocolates with high cocoa content and products considered to be pro-health, including cereal bars with additives such as fruit and nuts (Konsumpcja słodyczy w Polsce – raport. https://www.portalspozywczy.pl/slodycze-przekaski/wiadomosci/konsumpcja-slodyczy-w-polsce-raport,128961.html).
Polish consumers were selected for the study due to an apparent increase in interest in fruit and cereal bars as lower-calorie products compared to chocolate bars. The research project was financed from the Polish state budget. It was conducted in order to learn about consumer behavior and their expectations towards a segment that has the potential for further growth. Health-oriented trends related to food choices, including even the choice of confectionery, are observed worldwide. The data obtained in our study can be used by companies and food and nutrition experts in countries with similar economic development as Poland.
The Discussion and Conclusions have been modified following comments from reviewers.
Thank you very much for all your comments. We appreciate them very much. We hope that we were able to improve the article according to your suggestions. Thank you for taking the time to read and evaluate the text.

Reviewer 3 Report
The authors investigated the motives for choosing fruit and cereal bars among the Polish population and the association with participant characteristics. Although the topic may be potentially important in this field, there are several concerns, as mentioned below.
1) Abstract
The maximum word count of this section is 200 words.
2) Introduction
The authors should explain what is already known and unknown about the present theme in this section and state the necessity of the present study. However, descriptions of the background are not sufficient and irrelevant. Descriptions in lines 31-60 are unnecessary, whereas I do not understand why the authors focused on cereal bars. Do cereal bars the highest food sources of sugars intake in the Polish population? Further, descriptions in lines 61-63 should be referred to by appropriate rationales. Otherwise, these sentences were merely the authors' opinions. Further, the authors should explain why they targeted the Polish population. Does Poland have unique features compared to other European countries? The absence of studies (lines 70-71) is not a sufficient reason to conduct the present study.
3) Methods
The validity and reliability of the questionnaire used are unknown. The authors should provide sufficient rationale for the questionnaire or detail the procedure of development.
4) Results
This section is too long; the authors do not have to mention all results from Tables. The authors should reconsider what they have to show in this section. Additionally, it is a common mistake to split up into several Tables data that belong in one Table.
5) Discussion
This section should be started with a summary of the main results of this study, followed by their interpretation in light of known literature. Then, the authors should explain the importance of their results and what the present study adds to our knowledge on this subject. Therefore, descriptions in lines 329-333 should be deleted. There are many descriptions in this section which should be provided in the results section, and the authors should also discuss the limitation of this study. They also should revise the conclusion section.
Author Response

(The authors gave the same response as above.)
